# Low-Profile Broadband Dual-Polarized Dipole Antenna for Base Station Applications

**DOI:** 10.3390/s23125647

**Published:** 2023-06-16

**Authors:** Hao Feng, Mengyuan Li, Zhiyi Zhang, Jiahui Fu, Qunhao Zhang, Yulin Zhao

**Affiliations:** 1School of Electronics and Information Engineering, Harbin Institute of Technology, Harbin 150001, China; 2AVIC Research Institute for Special Structures of Aeronautical Composites, Jinan 250104, China; 3Department of Electrical Engineering, City University of Hong Kong, Hong Kong SAR, China; 4College of Metrology and Measurement Engineering, China Jiliang University, Hangzhou 314423, China

**Keywords:** dipole antenna, dual-polarized, low-profile, wideband artificial magnetic conductor

## Abstract

A low-profile broadband dual-polarized antenna is investigated for base station applications. It consists of two orthogonal dipoles, fork-shaped feeding lines, an artificial magnetic conductor (AMC), and parasitic strips. By utilizing the Brillouin dispersion diagram, the AMC is designed as the antenna reflector. It has a wide in-phase reflection bandwidth of 54.7% (1.54–2.70 GHz) and a surface-wave bound range of 0–2.65 GHz. This design effectively reduces the antenna profile by over 50% compared to traditional antennas without an AMC. For demonstration, a prototype is fabricated for 2G/3G/LTE base station applications. Good agreement between the simulations and measurements is observed. The measured −10-dB impedance bandwidth of our antenna is 55.4% (1.58–2.79 GHz), with a stable gain of 9.5 dBi and a high isolation of more than 30 dB across the impedance passband. As a result, this antenna is an excellent candidate for miniaturized base station antenna applications.

## 1. Introduction

As wireless communication networks expand, base station antennas are critical components in these systems. However, they face challenges, such as large physical dimensions and limited bandwidth [1]. Dual-polarized antennas have received great attention due to their advantages in polarization diversity and high isolation, which can increase channel capacity. Practically, various antenna designs can achieve dual polarization. Common elements can include patch antennas [2,3], slot antennas [4], or dipole antennas [5,6,7,8,9,10,11,12,13,14,15,16]. However, they may suffer from either a limited bandwidth [2,3,4] or a high profile [5,6,7]. Achieving miniaturization and widening the bandwidth have emerged as the foremost challenges in developing dual-polarized antennas.

Dual-polarized planar dipole antennas have attracted considerable attention because of their wide bandwidth and pattern stability. Low-profile designs can be achieved using bowtie-shaped crossed-dipole antennas and Huygens dipole antennas, with profiles of 0.088*λ*_0_ [8] and 0.0483*λ*_0_ [9], respectively (where *λ*_0_ represents the wavelength in free space at the center frequency). However, they can provide narrow −10-dB impedance bandwidths of only 15.6% [8] and 0.462% [9]. To enhance the antenna bandwidth, parasitic elements are used in antenna design. The common parasitic element can be L-shaped metal strips [10] or quadrangular loops [11,12]. They act as additional resonating elements, creating a coupling effect [10] or adding a new resonant mode [11,12]. As a result, the overall bandwidth significantly improved. The parasitic U-shaped grooves are also utilized to broaden the bandwidth [13,14]. Grooves can modify the current distribution on the main radiating element and, thus, improve the antenna impedance. However, the profile of those dipole designs is around a quarter wavelength due to the conventional conductor reflector. The profile is required to eliminate the effect of reflected electromagnetic waves from the reflector. This bulkiness increases the cost and may not be suitable for size-constrained applications.

Frequency Selective Surfaces have recently obtained much attention due to their unique properties in favor of antenna performance. A frequency selective surface (FSS) is a periodic structure, typically a thin sheet or a grid, that has unique transmission and reflection properties for different electromagnetic frequencies. These surfaces are designed to either pass or block specific frequency ranges when subject to electromagnetic radiation. The FSS is useful in antenna design, such as reducing decoupling [15,16] and increasing gain [17,18]. FSSs are usually located on the top of the antenna radiation aperture to realize some specific functions. This method is effective and straightforward. However, it will introduce extra antenna length, which may limit its practical applications.

As a kind of FSS, the artificial magnetic conductor (AMC) is useful in addressing antenna size problems. It can replace a traditional metal conductor as a new reflector type for base station antennas. They exhibit properties of magnetic conductors with a zero-degree phase shift upon reflection. The in-phase reflection avoids the 180-degree phase change of the electromagnetic wave when it is reflected from the metal plate, thus, eliminating the need to set the reflector back at 1/4 λ_0_ from the radiator. As a result, this characteristic enables them to reduce the antenna profile, leading to compact and low-profile antenna designs. In [8], using a circle-typed AMC, the profile of a dipole antenna is effectively reduced to 0.088*λ*_0._ However, its −10-dB impedance bandwidth is only 15.6% (2.36–2.76 GHz). In [19], using mushroom-type AMC, a patch antenna has an impedance bandwidth of 33% (24.05–33.52 GHz) with a profile of 0.053λ_0_.

Air gap technology has obtained much attention due to its ease of fabrication and high performance in antenna applications [20]. Adding an air gap in AMC can expand the in-phase bandwidth of AMC. In [21,22], dual-polarized dipole antennas using AMC reflectors have heights of 0.13λ_0_ and 0.15λ_0_. These antennas have a wide AMC bandwidth of more than 40%. However, both designs suffer from strong surface waves, leading to decreased antenna gain [21] or high cross-polarization at big angles [22]. Therefore, designing a wideband AMC that can suppress surface waves is highly desired.

In this paper, a low-profile broadband dual-polarized dipole antenna using a novel AMC reflector is investigated. Using the Brillouin dispersion diagram, the AMC is designed to suppress the surface-wave effect and decrease the antenna height. As a result, the proposed antenna has a high efficiency and low antenna profile. The fork-shaped lines feed the antenna without a direct connection, giving a broad impedance bandwidth. The parasitic strips reduce cross-polarization. The evolution of the proposed AMC and dipole antenna is analyzed using equivalent circuits and Smith charts. A prototype that operates in 2G/3G/LTE base station applications was designed and fabricated. The reflection coefficient, radiation pattern, antenna gain, and antenna efficiency are measured. Reasonable agreement between the measured and simulated results is obtained.

## 2. Antenna Design and Configuration

Figure 1 shows the configuration of the wideband dual-polarized antenna. With reference to Figure 1a,b, this antenna comprises crossed-dipoles, double-layer AMC, fork-shaped feeding structures, and parasitic strips. Two crossed dipoles are positioned perpendicularly with slant angles of *φ* = ±45° and are represented as dipoles 1 and 2. Figure 1c shows the exploded view of the proposed antenna. It can be seen from the figure that the feeding lines and crossed dipoles are printed on Layers 1 and 2 of Substrate 1 (Sub. 1), respectively. Figure 1d zooms in view of the arms of dipoles and feeding structures. As can be seen from the figure, each arm of the dipole is a hexagonal ring with a width of *w*_1_ = 3.6 mm and four mirror-symmetrical etched slots. It should be mentioned that one of the dipole arms Is soldered directly to the feeding structures, while the other arm is fed via space coupling. This method is designed to obtain a wide −15-dB impedance matching bandwidth. The novel AMC consists of two metal layers, one substrate, and air layers. Periodic patches are its first metal layer printed on Layer 3 of Sub. 2, with the interval gap of *g* = 2 mm and patch width of *w* = 13 mm. The second metal layer is aluminum ground. Sub. 1 and Sub. 2 are FR4 substrates (*ε_r_* = 4.4, tanδ = 0.02). Crossed dipoles are placed at a height of 5.5 mm from the AMC surface. Plastic pins are used to locate layers precisely. Figure 2 shows the different layers of the dual-polarized antenna. With reference to the figure, our parasitic strips are printed on Layer 2 of Substrate 1 to reduce cross-polarization.

## 3. Analysis and Discussion

### 3.1. In-Phase Reflection Bandwidth of the AMC Unit

Figure 3 shows the configurations and reflection coefficients of different AMC units. The proposed AMC is a patch-type with a patch length and width of *w* = 13 mm, a substrate-layer height of *h_d_* = 1.2 mm (0.0088*λ*_0_), and an air-layer height of *h_a_* = 12 mm (0.088*λ_0_*). AMC_1_ and AMC_2_, which are conventional AMCs with the same patch size and gap width but different substrate-layer heights (*h_d_*_1_ = 1.2 mm, *h_d_*_2_ = 9.5 mm), were simulated for comparison. Generally, the in-phase reflection bandwidth is the frequency range in which the reflection phase is between −π/2 and π/2. With reference to the figure, the in-phase reflection bandwidths are 9.8%, 39.4%, and 54.7% for AMC_1_ (4.57–5.04 GHz), AMC_2_ (1.67–2.49 GHz), and the proposed AMC (1.54–2.70 GHz), respectively. Compared with the proposed AMC, AMC_1_ lacks the air layer, giving a zero-phase reflection frequency of 4.82 GHz that is out of the operating bandwidth of 2G/3G/LTE. It is worth mentioning that AMC_2_ has almost the same zero-phase reflection frequency, however, it has a narrower in-phase bandwidth and a worse reflection coefficient than the proposed AMC. Therefore, our AMC is a good alternative to the metal ground due to its wideband. 

The proposed AMC unit can be equivalent to an LC parallel resonant circuit using the equivalent circuit analysis method (ECM) [23,24], as illustrated in Figure 4. Periodical patches provide the grid inductance *L_p_*. Another inductance *L_d_* is created by the total substrate, including the air and the dielectric. Gaps between adjacent patches form a grid capacitance *C_p_*. As a result, the resonant frequency and bandwidth of the proposed AMC can be represented by Equations (1) and (2), which show the relationship between parameters on the resonant frequency and bandwidth coefficient.
(1)$$f=\frac{1}{2\pi \sqrt{(L_{p}+L_{d})C_{p}}},$$
(2)$$BW=\frac{\pi }{8\eta_{0}}\sqrt{\frac{L_{p}+L_{d}}{C_{p}}}\times {\left(\frac{L_{d}}{L_{p}+L_{d}}\right)}^{2},$$
where $\eta_{0}=\sqrt{\mu_{0}/\varepsilon_{0}}$ is the wave impedance in free space. Compared with AMC_1_ and AMC_2,_ the proposed AMC has higher profiles and, thus, a larger dielectric inductance *L_d_*. Since the derivative of Equation (2) with respect to *L_d_* is positive, the bandwidth is positively related to the dielectric inductance *L_d_*. Therefore, AMC_2_ and the proposed AMC have a wider bandwidth. In addition, the dielectric substrate of AMC_2_ is thicker, which increases dielectric losses and manufacturing costs. The proposed AMC has the widest bandwidth. The surface impedance of AMC can be represented by: (3)$$\eta_{s}(f)=j2\pi fL_{d}\frac{1-{\left(2\pi f\right)}^{2}L_{p}C_{p}}{1-{\left(2\pi f\right)}^{2}(L_{p}+L_{d})C_{p}}.$$

At the parallel resonant frequency (using Equation (1)), the denominator of Equation (3) is equal to zero, and thus, the surface impedance *η_s_* tends to infinity. The reflection coefficient of AMC can be calculated by $\Gamma =(\eta_{s}-\eta_{0})/(\eta_{s}+\eta_{0})$, where $\eta_{0}$ equals the impedance of air. Therefore, the reflection coefficient equals one, and the reflection phase is zero. AMC achieves the in-phase reflection at the resonant frequency.

### 3.2. Brillouin Dispersion Diagram of the AMC Unit

Generally, a surface wave will be excited when electromagnetic waves reflect from the ground, deteriorating antenna performance. Figure 5 shows the dispersion curves of the proposed AMC unit. With reference to the figure, the electromagnetic wave under 2.65 GHz served as a slow wave and bounded in the AMC surface. However, as the frequency increases beyond 2.65 GHz, the electromagnetic wave gradually becomes a fast wave and leaks into free space. Consequently, antennas loading the proposed AMC and the conventional metal ground have nearly the same realized gain before 2.65 GHz, as shown in Figure 6a. Figure 6b shows the simulated S-parameters of the antenna. It can be seen from the figure that both the reflection coefficient and isolation are improved by using the AMC. It should be mentioned that the proposed antenna can effectively decrease the antenna profile from 0.25λ_0_ to 0.13λ.

### 3.3. Evolution of the Antenna

Figure 7 shows the design evolution of the crossed dipole and compares the simulated reflected coefficient of Dipole I, II, and III. Figure 8 provides electrical current distributions of different dipoles at their respective resonant frequencies. With reference to Figure 7, Dipole I is a conventional rectangular dipole with only one resonant mode (seen in Figure 8a), having a −10-dB impedance bandwidth of 29.9% (1.88–2.54 GHz). Dipole II is obtained by cutting the arm corners and centers of Dipole I, which introduces a new resonant mode *f_21_* in the lower-frequency band. Therefore, Dipole II has a wide −10-dB impedance bandwidth of 41.7% (1.67–2.55 GHz). This can be well-demonstrated by Figure 8b. The surface current distribution at *f_11_* and *f_23_* is the same.

Dipole III is obtained by adding rectangular slots on Dipole II. The introduced rectangular slots effectively make two different resonant modes close (seen in Figure 8c), resulting in a wide-impedance passband of 55.4% (1.58–2.79 GHz).

Figure 9 also shows the Smith charts of antennas with different dipoles. The Smith chart of Dipole I has no resonant loop, while the Smith charts of Dipole II and Dipole III have 1 and 2 resonant loops, respectively. Dipole III achieves a wider impedance bandwidth by cutting arm corners and centers and adding rectangular slots.

The equivalent circuit model of Dipole III is shown in Figure 10a. According to previous studies [25,26], the feeding structure is modeled as a parallel *LC* resonator circuit (*L_f_* and *C_f_*) and a π-shaped network (*C_g1_*, *C_g11_*, and *C_g12_*). Dipole III is obtained by cutting corners and opening rectangular slots in Dipole I. Dipole I can be expressed as a parallel *RLC* resonator circuit (*R_d1_*, *L_d1_*, and *C_d1_*). The corners and rectangular slots can be expressed as a parallel *RLC* resonator circuit (*R_s1_*, *L_s1_*, and *C_s1_*). When one loop dipole is excited, the other acts as a coupled loop resonator. Consequently, they are configured in parallel within the circuit. The adjacent gaps between two dipoles are modeled as a π-shaped network (*C_g2_*, *C_g21_*, and *C_g22_*).

By utilizing the proposed equivalent circuit and circuit analysis software, we can expedite the design process of the proposed antenna, particularly when attempting to bring two resonant modes into proximity. The values of circuit parameters can be obtained by using a curve-fitting method. The extracted results are *L_f_* = 1.73 nH, *C_f_* = 0.10 pF, *C_g1_* = 3.48 pF, *C_g11_* = 0.43 pF, *C_g12_* = 4.17 pF, *R_d1_* = 99.9 Ω, *L_d1_* = 1.07 nH, *C_d1_* = 0.36 pF, *R_s1_* = 5.02 Ω, *L_d1_* = 1.07 nH, *C_s1_* = 2.23 pF, *C_g2_* = 2.45 pF, *C_g21_* = 0.43 pF, *C_g22_* = 4.17 pF, *R_d2_* = 5.24 Ω, *L_d2_* = 0.35 nH, *C_d2_* = 3.18 pF, *R_s2_* = 5.24 Ω, *L_d2_* = 0.35 nH, *C_s2_* = 2.93 pF. Figure 10b shows the reflection coefficient derived from the equivalent circuit model and that from the HFSS.

AMC is employed to replace the traditional metal ground, thereby increasing the antenna gain and maintaining a low profile. Therefore, it becomes crucial to select an appropriate quantity of AMC units. This strategic selection aims to ensure that the antenna exhibits a broad, high-gain bandwidth, adequately covering the frequency ranges of 2G (1.71–1.92 GHz), 3G (1.88–2.17 GHz), and LTE (2.3–2.4 GHz and 2.5–2.69 GHz) networks. Figure 11 shows the simulated realized gains of the proposed antenna with different numbers (*m × m*) of AMC units. With reference to Figure 11, when the number of elements is 11 × 11, the proposed antenna has the highest average gain across the bandwidth of 1.71–2.69 GHz. Therefore, the number 11 × 11 is chosen for AMC design.

Figure 12 shows the comparison of simulated radiation patterns between the crossed-dipole antenna with/without four parasitic strips. With reference to the figure, significant improvements in both *E*- and *H*-plane radiation patterns can be obtained by the loading strips. It is worth mentioning that, using this method, the cross-polar fields in the two principal cutting planes decrease by more than 12 dB.

## 4. Results

To verify the idea, a prototype of the proposed antenna was fabricated, as shown in Figure 13. Figure 14 shows the measured and simulated S-parameter of our antenna. With reference to Figure 14a, the measured and simulated −10-dB impedance bandwidths (|S11| ≤ −10 dB) are 54.05% (1.62–2.82 GHz) and 55.37% (1.58–2.79 GHz), respectively. As shown in Figure 14b, the isolations in both the simulation and measurement are almost greater than 30 dB across the impedance-matching passband. As can be observed from Figure 14, reasonable agreement between the measured and simulated results is observed. Acceptable errors are caused by slight deformation due to uneven structural support. The designed antenna exhibits good broadband characteristics and high isolation.

Figure 15a presents simulated and measured realized gains of the proposed antenna at *θ* = 0°. With reference to the figure, reasonable agreement can be found between the simulated and measured results. The ripple in measurement is less than 1 dB caused by experimental imperfections. The measured gain varies between 8.3 and 10.8 dB over the frequency range (1.62–2.82 GHz) shown in Figure 15a. Figure 15b shows the measured antenna efficiency of the low-profile crossed-dipole antenna. As can be observed from Figure 15b, the measured antenna efficiency is higher than 90% across the entire operating band.

Figure 16 shows the measured and simulated normalized radiation patterns in two principal cutting planes. Reasonable agreement can be found between simulations and measurements. As shown in the figure, stable radiation patterns can be found at three different frequencies. The simulated cross-polar field is almost lower than −30 dB. It can be observed from the figure that the co-polar field is 10 dB higher than the cross-polar field over a wide range of angles (−60° < *θ* < 60°) in the measurements. Results of Port 1 are only shown due to the geometric symmetry of our crossed-dipole antenna.

Table 1 compares our antenna with the reported crossed-dipole antennas. With reference to the table, the antennas in [8,9] have an extremely low antenna height but a narrow bandwidth and low gain. The designs in references [11,12,13,14] have a higher profile. The antennas in references [14,21] have similar profiles, but their antenna gains are lower than the proposed antenna. Our wideband dual-polarized antenna has the highest antenna efficiency due to the proposed AMC. Compared with listed base-station antennas, the proposed antenna outperforms in overall performance. It is a good candidate for base-station applications.

## 5. Conclusions

A wideband low-profile crossed-dipole antenna with a novel AMC has been investigated. By introducing the AMC as antenna ground, the profile can be effectively decreased to 0.12λ_0_. In addition, four parasitic metal strips have been introduced to reduce the cross-polar field and, thus, increase the antenna gain in its high-frequency impedance passband. To verify the idea, a prototype has been simulated, fabricated, and measured. A measured overlapping bandwidth of 55.4% (1.58–2.79 GHz) has been obtained. Its maximum measured realized gain is 10.8 dB at 2.6 GHz. Additionally, a stable gain and radiation pattern have been observed. The prototype has a measured isolation of ~30 dB. It should be mentioned that its cross-polarization level is more than −10 dB over a wide range of angles (−60° < *θ* < 60°). Finally, these advantages enable the proposed antenna to be potentially applied to the integrated design of 2G/3G/LTE base station antennas.

## Figures and Tables

**Figure 1 sensors-23-05647-f001:**
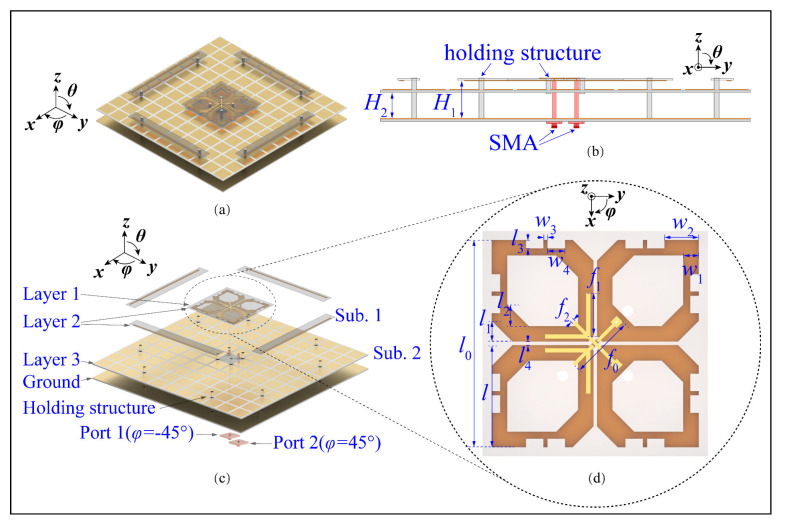
Configuration of the dual-polarized antenna. (**a**) Perspective view, (**b**) side view, (**c**) exploded view, and (**d**) zoomed view of crossed-dipole antenna. *H*_1_ = 17.5 mm, *H*_2_ = 12 mm, *w*_1_ = 3.6 mm, *w*_2_ = 7.3 mm, *w*_3_ = 1.65 mm, *w*_4_ = 4 mm, *l* = 23.5 mm, *l*_0_ = 52 mm, *l*_1_ = 4.8 mm, *l*_2_ = 5 mm, *l*_3_= 2 mm, *l*_4_ = 0.8 mm, *f*_0_ = 14 mm, *f*_1_ = 10.5 mm, *f*_2_ = 0.83 mm.

**Figure 2 sensors-23-05647-f002:**
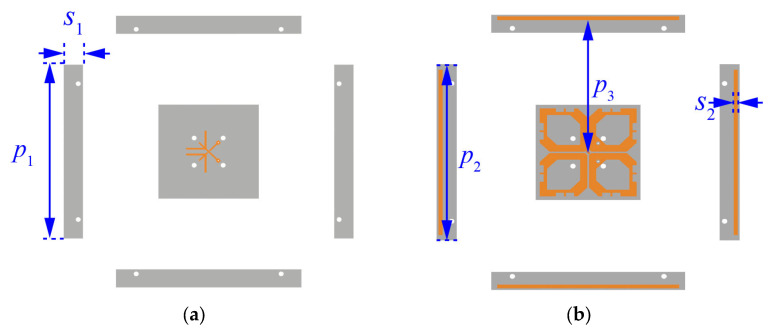
Different layers of the dual-polarized antenna: (**a**) Substrate 1 Layer 1, (**b**) Substrate 1 Layer 2, (**c**) Substrate 2 Layer 3, and (**d**) aluminum ground. *l*_5_ = 65 mm, *w* = 13 mm, *g* = 2 mm, *S*_1_ = 10 mm, *S*_2_ = 2 mm, *P*_1_ = 96 mm, *P*_2_ = 90 mm, *P*_3_ = 72 mm.

**Figure 3 sensors-23-05647-f003:**
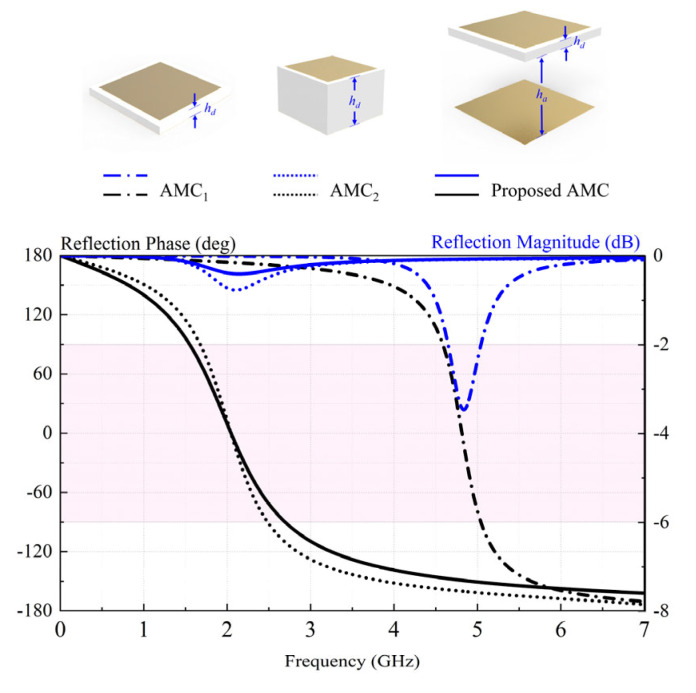
Reflection coefficients and configurations of different AMC units. The pink area is the in-phase reflection bandwidth.

**Figure 4 sensors-23-05647-f004:**
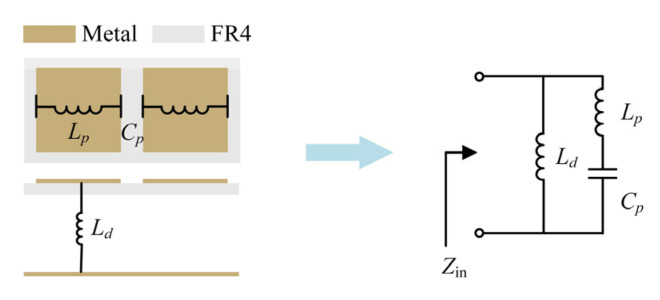
Equivalent circuit model of the proposed AMC unit.

**Figure 5 sensors-23-05647-f005:**
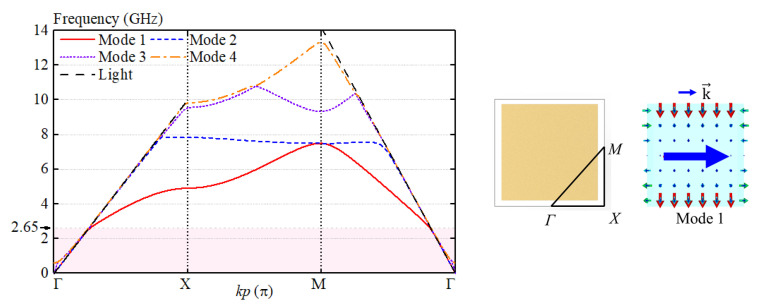
The dispersion curves of the proposed AMC unit. The illustrations are the electric field distributions. The blue arrow indicates the direction of propagating wave.

**Figure 6 sensors-23-05647-f006:**
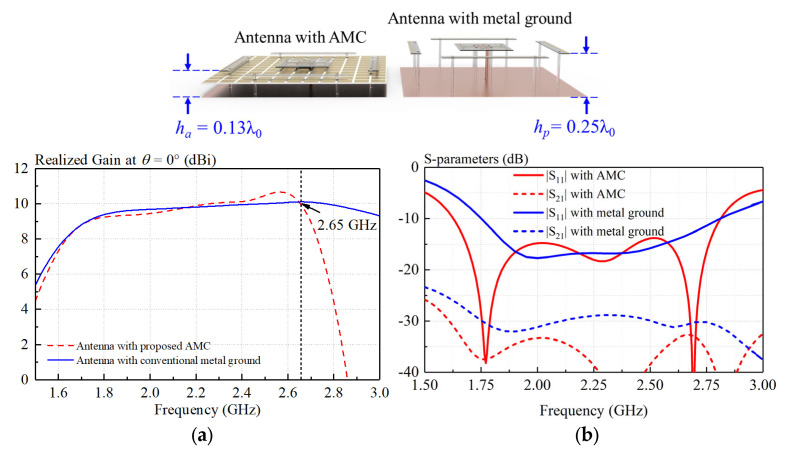
Comparison between crossed-dipole antenna with AMC and that with conventional metal ground. (**a**) Simulated realized gain, (**b**) simulated S-parameters.

**Figure 7 sensors-23-05647-f007:**
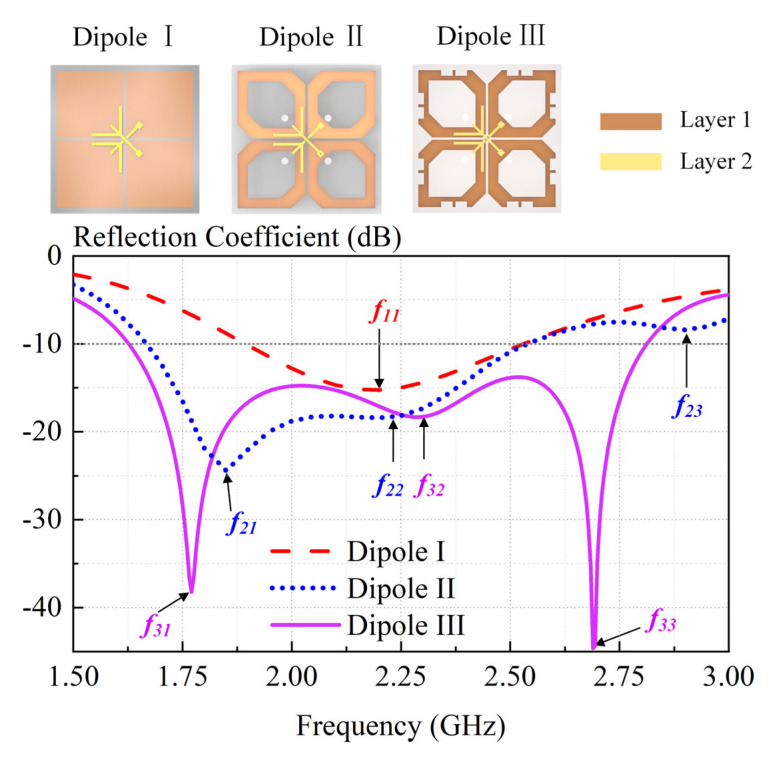
Schematic diagrams and simulated reflection coefficients of Dipole I, Dipole II, and Dipole III. Dipole I (a conventional crossed-dipole antenna), Dipole II (Dipole I with corners and centers cut), and Dipole III (crossed dipole in proposed antenna).

**Figure 8 sensors-23-05647-f008:**
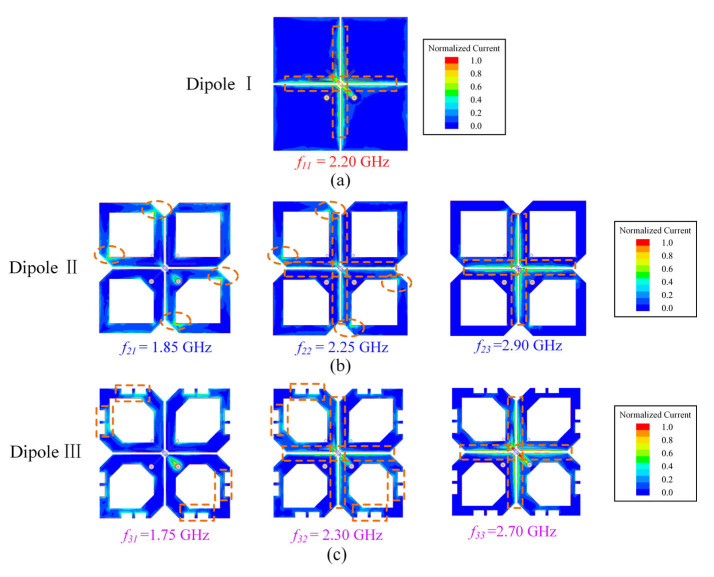
Electrical current distributions of (**a**) Dipole I, (**b**) Dipole II, and (**c**) Dipole III.

**Figure 9 sensors-23-05647-f009:**
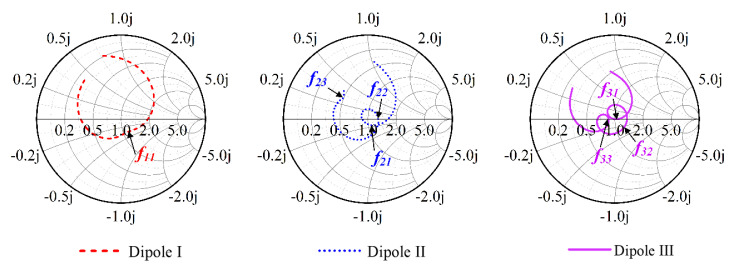
Smith charts of antennas of Dipole I, Dipole II, and Dipole III.

**Figure 10 sensors-23-05647-f010:**
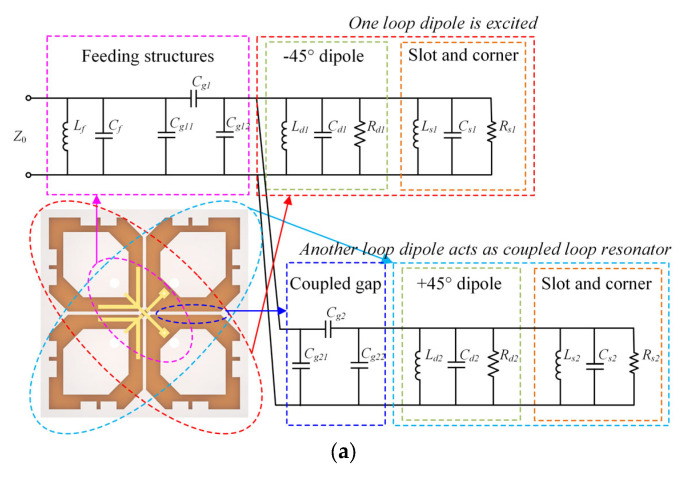
(**a**) Equivalent circuit model of Dipole III. (**b**) Reflection coefficient of Dipole III obtained from the equivalent circuit model and that from the HFSS.

**Figure 11 sensors-23-05647-f011:**
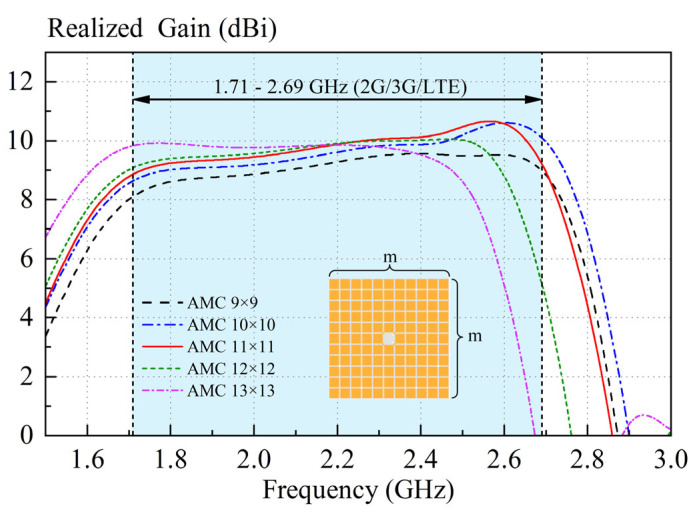
Simulated realized gain for proposed antenna under different numbers (*m × m*) of AMC units. The blue region is the commercial bands for 2G, 3G, and LTE.

**Figure 12 sensors-23-05647-f012:**
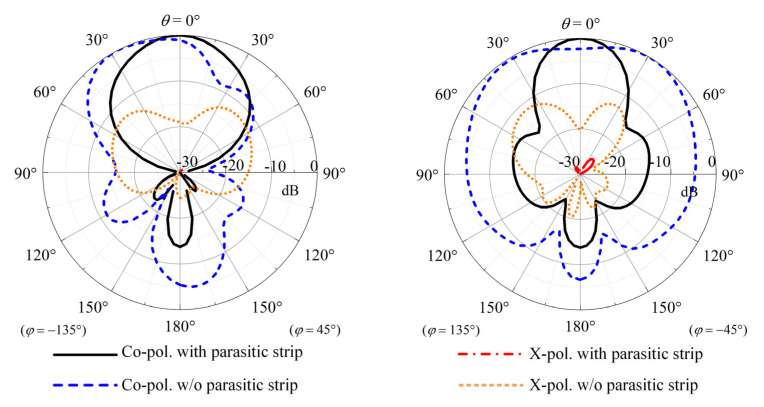
Comparison of simulated radiation patterns between the antennas with/without parasitic strips.

**Figure 13 sensors-23-05647-f013:**
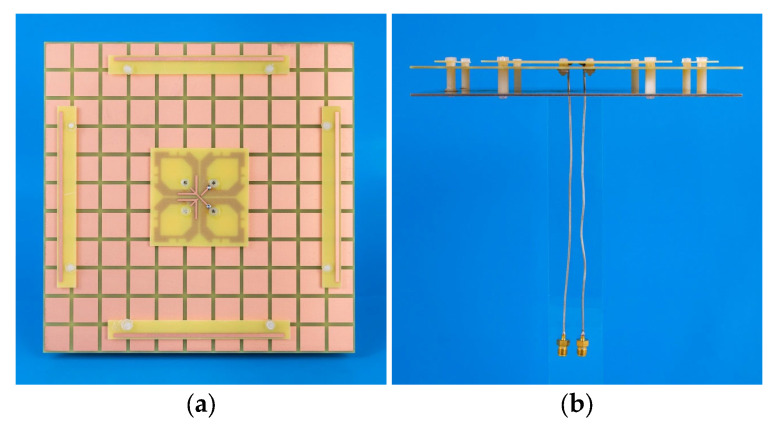
Photograph of proposed antenna. (**a**) Top view, (**b**) side view.

**Figure 14 sensors-23-05647-f014:**
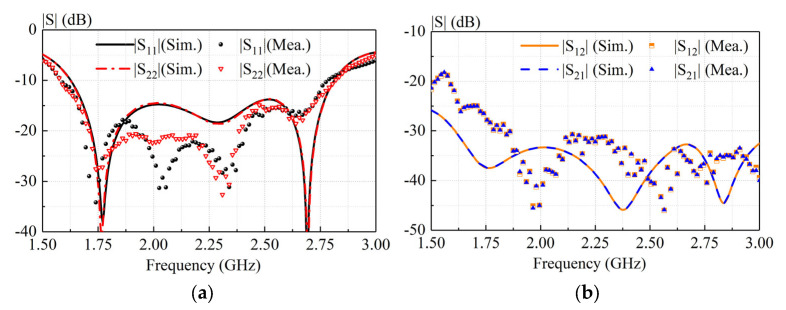
Measured and simulated S-parameters of proposed antenna.

**Figure 15 sensors-23-05647-f015:**
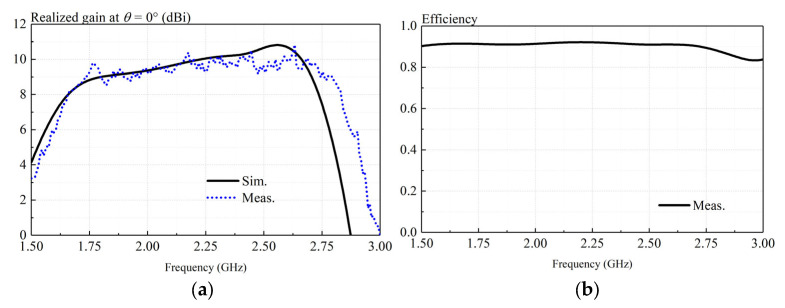
Measurements and simulations of proposed antenna. (**a**) Realized gain; (**b**)antenna efficiency.

**Figure 16 sensors-23-05647-f016:**
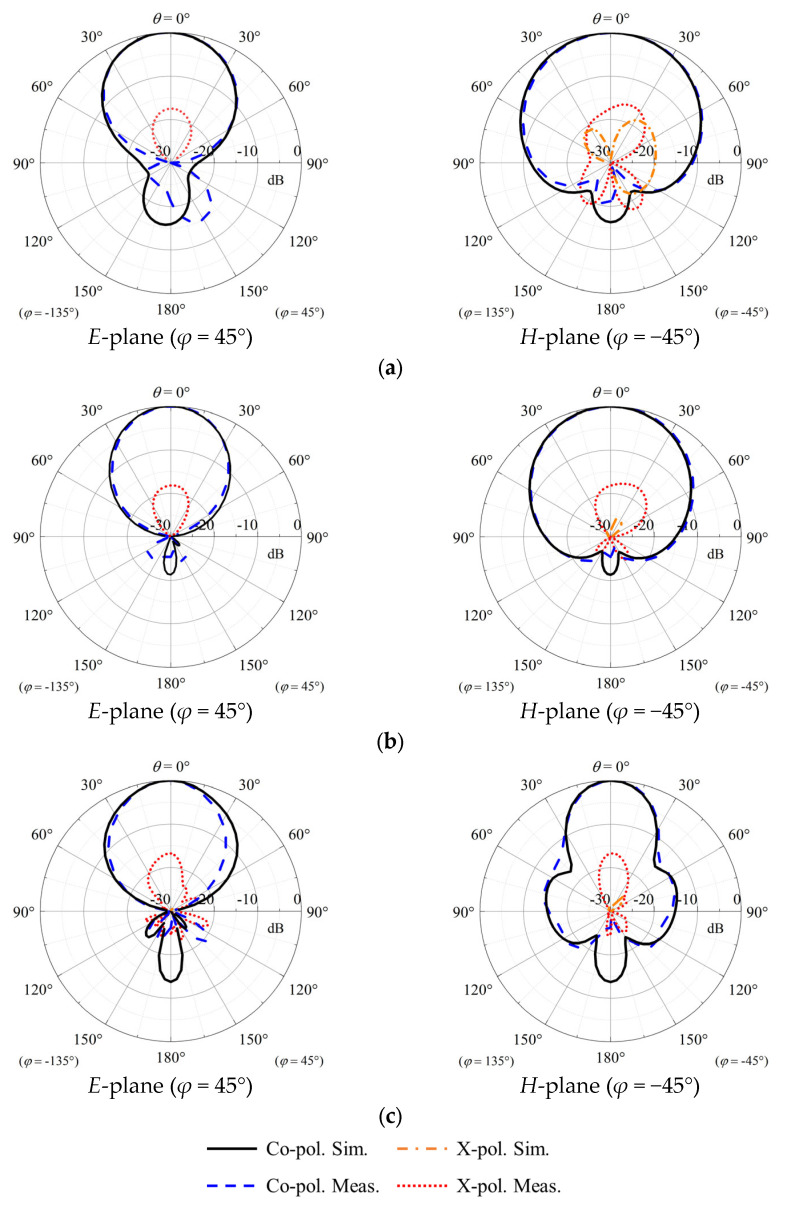
Measured and simulated radiation patterns of proposed antenna Port 1. (**a**) 1.6 GHz, (**b**) 2.1 GHz, and (**c**) 2.6 GHz. Results of Port 1 are only shown due to the geometric symmetry of our crossed-dipole antenna.

**Table 1 sensors-23-05647-t001:** Comparison of crossed-dipole antennas.

Ref.	−10-dB Impedance Bandwidth	Gain (dBi)	Isolation (dB)	Antenna Efficiency	Profile (λ_0_)
[8]	15.6% (2.36–2.76 GHz)	4.6–7.2	>22	65.0%	0.09
[9]	0.46% (1.48–1.55 GHz)	<2.2	>26	61.3%	0.05
[11]	52.2% (1.70–2.90 GHz) (VSWR < 1.5)	8.2–9.4	>26	N.A.	0.26
[12]	48.7% (1.66–2.73 GHz)	7.8–8.5	>34	85.0%	0.23
[13]	65.3% (1.67–3.29 GHz)	7.8–8.8	>32	90%	0.26
[14]	74.9% (1.31–2.88 GHz)	7.8~9.1	>30	N.A.	0.35
[21]	56.3% (1.67–2.98 GHz)	6.7–7.6	>25	80%	0.13
[22]	49.4% (1.69–2.80 GHz)	6.8–9.8	>27	86%	0.15
This work	55.4% (1.58–2.79 GHz)	8.3–10.8	>30	90%	0.13

## Data Availability

Not applicable.

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
