# Peer review of "Low-Profile Broadband Dual-Polarized Dipole Antenna for Base Station Applications"

_sensors, 2023, doi:10.3390/s23125647_

Round 1

Reviewer 1 Report

The authors presented Low-Profile Broadband Dual-Polarized Dipole Antenna for Base Station Applications. The concept is exciting, and the simulation results are reasonably good, showing strong reconfigurability. I have the following suggestions before accepting it for publication:

First, the introduction needs improvement. In my opinion, a total of 16 references is not sufficient. Additionally, since the authors utilized the AMC as a reflector to enhance the gain, there should be an explanation of the techniques of Metamaterials used to achieve this, such as the implementation of frequency selective surface (FSS) [1-4]. I recommend considering these articles, as they may add value to the introduction.

[1] Compact Size and High Gain of CPW-Fed UWB Strawberry Artistic Shaped Printed Monopole Antennas Using FSS Single Layer Reflector," in IEEE Access, vol. 8, pp. 92697-92707, 2020, doi: 10.1109/ACCESS.2020.2995069.

[2] Design of Broadband High-Gain Fabry–Pérot Antenna Using Frequency-Selective Surface. Sensors 202222, 9698. https://doi.org/10.3390/s22249698.

[3] High gain antenna at 915 MHz for off-grid wireless networks," Bulletin of Electrical Engineering and Informatics (BEEI), vol. 9, no. 6, pp. 2449–2454, 2020, doi:10.11591/eei.v9i6.2192.2020.

[4] Circuit Modelling of Broadband Antenna Using Vector Fitting and Foster Form Approaches for IoT Applications. Electronics 202211, 3724. https://doi.org/10.3390/electronics11223724.

Second: Please do not present Figure 1 without providing an explanation of the figure beforehand.

Third: The authors discussed the dimensions of the AMC unit cell in terms of width and height, but they did not mention the length. Please clarify the length of the unit cell as well.

Fourth: Why did the authors achieve an in-phase reflection? This is an important question regarding the enhancement of antenna engineering.

Fifth: The authors should mention the number of unit cells used in the AMC, which is 11 times 11, resulting in a total of 121-unit cells. It is important to explain why this specific number of unit cells was chosen and why alternatives such as 10 times 10, 9 times 9, 8 times 8, and so on, were not selected.

Sixth: It would be highly appreciated if the authors could provide the equivalent circuit model (ECM) of the dipole antenna design (3).

Seventh: Please add another column to Table 1 and compare it with the existing literature work.

Addressing these comments and concerns will greatly enhance the quality and impact of the research. 

That's all for me at this moment! The authors are required to revise the comments above carefully. Thanks

There are minor errors that need to be carefully checked!

Author Response

Dear Editorial Board,

Many thanks for your time and efforts in handling our manuscript.

We thank the reviewers for their careful review and constructive comments. The manuscript has been revised according to the comments of the reviewers. Please refer to our replies to the reviewers and editors for details in the attachment.

We look forward to receiving your early reply.

Best regards,

Zhi-Yi ZHANG

City University of Hong Kong
83 Tat Chee Avenue, Kowloon Tong
Kowloon Tong
Hong Kong

E-mail: zhiyi.zhang@my.cityu.edu.hk

Reviewer 2 Report

Authors should improve technical writing of the paper.

Minor editing of English language is required.

Author Response

(The authors gave the same response as above.)

Reviewer 3 Report

The paper concerns the design of a novel dual-pol dipole antenna for base station applications. The thickness reduction and gain increase are achieved by using the AMC substrate. The antenna bandwidth is enhanced with the edge slots on the dipole arms. Polarisation isolation is improved with the aid of parasitic passive scatterers. Although each of these techniques has been extensively studied and used in the similar context previously, their combination seems to be novel and, apparently, better in some aspects than existing solutions. Paper is written well and contains essential details of simulations and measurements. Apart from just a couple of typos, no language revision is required. Our major comment is that the depth of the analysis and discussions might be improved, possibly by means of Smith charts and equivalent circuits to clarify critical aspects of the design. The evolution of the designs is lacking in-depth insights. E.g., whilst reading about the AMC development from AMC1 to AMC3, one would naturally wonder what was the actual physical mechanism behind the BW improvement (this can be readily demonstrated with a Smith Chart). In the absence of deeper insight into the physical mechanisms underlying antenna design, the proposed development may look in some aspects incremental, as compared to the state of the art.  

Author Response

(The authors gave the same response as above.)

Round 2

Reviewer 1 Report

Dear authors,

Thank you for taking the time to revise the paper carefully. However, I believe that the article is now ready to be published in its current form.

There are minor errors that need to be carefully checked!